# Empirical Study on Households’ Energy-Conservation Behavior of Jiangsu Province in China: The Role of Policies and Behavior Results

**DOI:** 10.3390/ijerph16060939

**Published:** 2019-03-15

**Authors:** Ting Yue, Ruyin Long, Junli Liu, Haiwen Liu, Hong Chen

**Affiliations:** School of Management, China University of Mining and Technology, Xuzhou 221116, China; TS18070114A31@cumt.edu.cn (J.L.); TS18070113A31@cumt.edu.cn (H.L.); hongchenxz@cumt.edu.cn (H.C.)

**Keywords:** household, energy-conservation behavior, behavior results, contextual factors

## Abstract

With the improvement of living quality and the increase of energy consumption of residents, their energy conservation behavior (ECB) plays an increasingly important role in energy conservation and emission reduction. As a kind of environmental behavior, ECB of residents is a complicated process. In this paper, ECB is divided into four types, considering habit adjustment, quality threshold, efficiency investment, and interpersonal facilitation. A comprehensive conceptual framework is built, adding perception about energy conservation results (PER) and contextual factors from a new perspective. Based on a survey in Jiangsu province of China, this paper examines the impact of intention on behavior under the moderation of contextual factors, as well as the effect of perception of energy-conservation results on intention and ECB by means of multivariate statistical analysis. The results show that the intention of energy conservation is the determinant of behavior, but it does not well transform into behavior, especially into quality threshold and interpersonal facilitation behavior. Different contextual factors have positive effects on the relationship of intention and different behavior. However, modulating effects of contextual factors as amplifiers do not function effectively due to their low rating scores. PER has a positive impact on intention but not on all types of ECB. Finally, this paper presents important implications for policy makers to optimize energy conservation policy.

## 1. Introduction

Global climate control has recently emerged as one of the most important international issues of the early 21st century. The large amount of carbon dioxide emissions caused by the rapid growth of fossil energy consumption is one of the main causes of global warming [1]. Therefore, to mitigate climate change, an emphasis on diverse actions across sector boundaries is required, and a reduction of energy consumption is urgently needed to limit global warming [2]. Because households are responsible for a considerable amount of the total greenhouse gas emissions [3,4], changes in their behavior to reduce energy consumption is recognized as part of the mitigating actions that are required to reduce emissions. In China, with the rapid development of the economy and society over the past decades, households’ energy consumption has sharply increased [5]. As of 2016, residential direct-life energy consumption accounted for approximately 12.44% of the total energy consumption in China, which has almost doubled since 2005 [6]. Based on the experience of the developed countries, the percentage of household energy use in total energy consumption will increase continuously with the development of the economy and society [7]. An individual resident is the most basic energy consumer, and households are the terminal link of energy consumption. Their behaviors not only directly affect the scale and growth of energy consumption and carbon emissions but also drive the emissions of industry, construction, transportation, and the service industries [8,9]. Therefore, the household potential for reduction is of great significance for environmental conservation. 

In most of the studies of energy conservation behavior (ECB), the classical environmental behavior models are always the theoretical basis in which psychological characteristics and contextual factors are usually key variables [10,11,12]. When conducting research on attitudes of energy conservation, a gap between intention and ECB is found [13]. This indicates that individuals seem to have green attitudes but have few actual pro-environmental behaviors. The reason may be that people pay more attention to comfort and convenience. The gap between green travel attitudes and behaviors in China is also confirmed [14]. China has a special background of “high-context culture”. The face sometimes means an individual’s public image in society, which involves concerns of social status and reputation. Face consciousness and conspicuous consumption usually result in high consumption [15,16], an individual behavior that is not only affected by psychological factors but by contextual factors as well [17]. Therefore, in the high-context culture of China, whether there is inconsistency between intentions and residents’ ECB is worthy of study. Between the intention and behavior, the contextual factors that correspond to moderating functions cannot be ignored. Additionally, perception about energy conservation results (PER) has an impact on behavioral intention [18,19]. Furthermore, in the case of inconsistency between intention and behavior, the effect of behavior results are more important. 

The purposes of this study are listed below: (1)To explore whether there is any inconsistency between intention and ECB.(2)To test whether there is a moderating effect of contextual factors.(3)To understand if different PER have the same effect on intention and behavior.

To achieve these purposes, this study classified ECB, contextual factors, and PER, built a conceptual framework, and proposed hypotheses. Then, empirical research and statistical analysis and discussion were carried out. The rest of this paper is arranged as follows. Section 2 presents the literature review. In Section 3, a conceptual framework of behavioral theories and hypotheses is presented. Section 4 presents the survey design, experimental procedures, and sampling strategy. In Section 5, the results of the empirical analysis are presented and discussed. Finally, in Section 6, conclusions are summarized, and policy implications are discussed.

## 2. Literature Review

A great deal of research has explored the influential factors of ECB. Additionally, the factors of residents’ psychological characteristics and intentions have traditionally been important factors in the formulation of energy policy [3,20]. For instance, a positive attitude towards the environment is conducive to family investment in energy efficiency [21]. Sütterlin et al. (2011) carried out the cluster analysis based on energy-related psychosocial factors and classified energy consumer into six segments [22]. Most psychological factors do not directly play a role in behavior but rather through intention. Intention is the indispensable process of behavior formation and the determinants that occur before the implementation of behavior [23]. Therefore, the intermediary roles of behavior and intention should be included in the study of ECB. 

Individual behavior is also affected by contextual factors [24,25]. The contextual factors such as energy conservation policies, price, the behavior of others, the provision of information about energy usage, culture, moral rules, and other social norms can strengthen intention of ECB [26,27,28,29]. Gärling et al. (2003) found that, when compared with price factors, pressure from the public had more obvious and durable effects on ECB [30]. The social norms of perceived environmental stress are particularly significant to ECB in the high-context and collectivist culture in China [7,31]. Studies also show that attributes of energy conservation products such as credibility, quality, and availability are important contextual factors [32,33]. Traditionally, the price of energy and energy efficiency production, subsidies of energy efficiency production, tax preference, rewards, and punishments are common economic policies [34], but economic measures without long-term incentive effects are often considered to be unfair and easy to abuse. Thus, non-price measures such as encouragement, persuasion, and information provision, which are considered to be low-cost and effect-lasting measures, have gained increasing attention [35,36,37]. 

In addition to the psychological and contextual factors, the outcomes of ECB are also important. Ajzen (2002) conducted a study on residual effects of behavioral outcomes and found that the residual impact of past behavior exists but vanishes when intentions are well-formed [18]. Individuals sometimes misperceive the effectiveness of potential ECB or overestimate the benefit of ECB. Once they learn about the deviations between perception and actual results, their behavior changes [38,39]. As a consequence, PER is also vital to intention and behavior [19,40]. 

Overall, there have been few surveys of residents’ ECB that consider the inconsistency between intention and behavior, the modulatory role of contextual factors, and the effect of PER in one study. Therefore, in this study, we chose the urban households of Jiangsu province as respondents to examine the effect of contextual factors, the inconsistency between intention and actual behavior, and the effect of PER. 

## 3. Hypotheses and Conceptual Framework

### 3.1. Conceptual Framework

Three classical theory models are often used to build models of ECB. The theory of planned behavior proposed by Ajzen (1991) is the succession and expansion of the theory of reasoned action. This theory believes that behavioral intention is the intervening variable between psychological characteristics and behavior. Individual behavior is not only affected by one’s own psychological characteristics but also by the surrounding environment [10,41]. The theory model of responsible environmental behavior proposed by Hines et al. (1987) shows that in the process of behavioral implementation, situational factors have a significant influence on the implementation of behaviors [11]. The theory of interpersonal behavior constructed by Triandis and Harry (1979) indicates that intention and external factors jointly influence behaviors. This theory also focuses on the influence of social factors and external factors on behavior [12].

According to Ajzen’s study about the residual effects of behavioral outcomes, the residual effect of behavior results does exist. When the behavioral intention is consistent with the actual behavior, the residual effect is small. When the behavioral intention is very strong, the expected realistic behavior is bound to be realized, and there is a specific behavioral intention implementation plan, the residual effect no longer exists [18]. 

This study selected the situational variables and classified ECB and the perception of PER with reference to the relevant literatures above as well as expert consultation and in-depth interviews based on the grounded theory. The variables used in the text are defined as shown in Table 1.

In this study, residents’ ECB was divided into four types: (1) habit adjustment behavior (HAB): turning off the lights when one leaves the room and reducing standby power consumption, and so on; (2) quality threshold behavior (QTB): buying fewer or using fewer electrical appliances, using air conditioning and heating in moderation, and so on; (3) efficiency investment behavior (EIB): choosing energy conservation types while buying lamps and home appliances, and so on; (4) interpersonal facilitation behavior (IFB): advising friends, families, or colleagues about how to save energy, sharing energy saving experiences, taking the initiative to prevent others’ energy wasting behavior, actively participating in energy conservation and the environmental protection community, and so on. Contextual factors include social norms of energy conservation (SNEC), popularization of energy conservation policy (PEP), execution and validity of energy conservation policy (EVEP), execution and validity of information intervention (EVII), price of energy (PRIE), energy conservation product attributes (EPA), and price of energy conservation products (PRIP). Perception about energy conservation results contains perception about energy conservation results on economic savings (PERE) and perception about energy conservation results in spiritual satisfaction (PERS).

The conceptual framework of residents’ ECB was built as shown in Figure 1.

### 3.2. Hypotheses

The behavior of an individual is not only affected by intention but is also influenced by the surrounding environment and other individuals’ behaviors. Because PER is also vital to subsequent behavior and intention [18,38,39,40], the influence of contextual factors and PER should be considered. Therefore, this paper mainly discusses four acting paths: (1) the acting path of intention on ECB; (2) the modulatory effect of contextual factors on the path of intention to behavior; (3) the effect of PER on behavior intention of energy conservation (BIEC); (4) the effect of PER on behavior.

On the basis of the theory of planned behavior [10,41] and the theory model of responsible environmental behavior [11], there is a significant positive relationship between behavioral intention and behavior. Behavioral intention is the direct antecedent variable of realistic behavior, and other subjective psychological factors all indirectly affect actual behavior through behavioral intention. The first hypothesis is put forward as follows:
**H1.** BIEC has positive effect on all of the four behaviors.

Individual behavior is also affected by external factors, which are otherwise referred to as contextual factors [17,37]. ECB can be modulated by contextual factors such as energy conservation policies, the behavior of others, and the provision of information about energy usage [26]. According to the review of the contextual factors above, the second hypothesis is proposed as follows:
**H2.** Contextual factors have significant modulatory effects on the path of intention to the four types of behavior.

Ajzen (2002) confirmed that behavioral outcomes can act on intentions through antecedent variables of intentions, and behavioral outcomes have a direct impact on behaviors [18]. Additionally, based on the related literature in the literature review, the third and the fourth hypotheses are put forward as follows:**H3.** The two types of PER have significant impacts on BIEC.
**H4.** The two types of PER have significant impacts on the four types of ECB.

## 4. Methods

### 4.1. Survey Design

In reference to the maturity scale [9], this paper modified and improved the scale items based on the related literature, expert consultation, and in-depth interviews with residents. The scales of the questionnaires contained socio-demographic characteristics, ECB, behavior intention, PER, and contextual factors evaluated by the Likert 5 method. Using ECB as an example, examples of four types of energy conservation behavior were presented (e.g., turning off the light when leaving a room). The answers were given on a five-point scale: rarely, sometimes, half the time, mostly, and very often.

To test the reliability and validity of the scales and the rationality of the length and expression of the questionnaire, a small preliminary survey was addressed through a network questionnaire survey after the establishment of the preliminary questionnaire scale. According to the results of the preliminary survey, ambiguous and vague items were modified or deleted to ensure the content validity of the questionnaire scale. On the basis of the results of the reliability test analysis, the items whose values of Cronbach’s alpha were under 0.6 were modified until their values increased up to 0.6, ensuring the scales could be accepted. Based on the exploratory factor analysis and confirmatory factor analysis, the items for which the factor loadings were under 0.5 were also deleted, and the formal investigation questionnaire scales were set.

### 4.2. Procedure and Sample

The combined survey methods of paper and network were used to make up for the inadequacy of the network survey and to obtain a uniform distribution of the demographic characteristics. The formal investigation began on 1 January 2018 in Jiangsu province of China. Until 25 February 2018, a total of 236 paper questionnaires and 478 network questionnaires were recycled. Eventually, there were 195 valid print questionnaires and 368 online effective questionnaires for a total of 563. The effective rate was 78.85%. The valid samples were evenly distributed in social demographic variables, and the samples were representative. Table 2 describes the general characteristics of the respondents. From the statistical description of the demographic characteristics, it can be seen that the sample distribution was uniform and, to a great extent, agreed with the current social situation.

## 5. Results and Discussion

### 5.1. Test of Scales and Descriptive Statistical Analysis of Variables

The results of the reliability and validity tests (Table 3) showed that the scales used in the questionnaires had high internal consistency and reliability, as indicated by a Cronbach’s alpha. The results of the Kaiser Meyer Olkin (KMO) measure, the chi-square values of Bartlett’s test of sphericity, and significance testing showed that this questionnaire was suitable for factor analysis [42]. The results of the exploratory factor analysis showed that the extracted factors corresponded to the variables of the conceptual framework; therefore, the scales had a high conceptual validity. On the basis of the maximum likelihood estimate method, the normality of the questionnaire survey data was tested, and the results showed that the kurtosis and skewness coefficients conformed to the normality requirement. Thus, related statistical analysis was permissible.

Through descriptive statistical analysis, ECB and its influencing factors could be well understood. The descriptive statistics results of the scale factors are shown in Table 3.

#### 5.1.1. Descriptive Statistics Analysis of Energy-Conservation Behavior

As is shown in Table 3, HAB was better implemented than the others, while IFB had the lowest score. As for HAB, it can be seen that the respondents had some energy conservation habits in daily life, such as “turning off lights when leaving”, and half of the respondents noticed the details of energy use such as “opening the refrigerator door as little as possible” and “adjusting or closing the gas valve in time”, but “when the appliance is not in use, cut off the power in time” was poorly implemented. On the whole, respondents had some habits of energy conservation in daily life. However, certain details that are not as easily attained (such as standby energy consumption) were always ignored. About half of the respondents were not familiar with the energy conservation knowledge that standby energy consumption causes a big waste of power. It shows that residents’ knowledge of energy conservation in daily life is not comprehensive enough. 

In regard to QTB, life quality was more important than energy conservation for most of the respondents. Most of them were unwilling to limit their quality of life in ways such as “using air conditioning in moderation”, “shortening bath time”, or “choosing green travel”. In terms of daily energy use related to quality of life, the majority of respondents seldom reduced the quality of life to save energy. This was consistent with the conclusion of Valkila and Saari (2013) [13]. People know how to reduce their energy use but are too comfort-loving to make any changes to their energy use. Therefore, advocating energy conservation should try to satisfy residents’ everyday needs and avoid lowering the quality of residents’ lives [43].

As for EIB, most of the respondents gave priority to “efficient bulbs and appliances” when buying household appliances, but few of them regarded fuel efficient or small displacement vehicles as their first choice. On account of the promotion of energy saving lamps and efficient appliances by the Chinese government in recent years, most residents have been able to give priority to energy saving products when choosing necessities of life. However, when buying household cars, the majority of respondents considered safety first, then comfort. At present, there are still barriers in the uptake of electric vehicles, such as low performance of electric vehicles and inconvenient charging [44]. 

In regard to IFB, a few of the respondents took the initiative to prevent energy waste of others or to share their experiences of energy conservation. Similarly, they rarely took part in activities, community, or energy conservation organizations. In China’s high-context culture, people’s consumption behavior is significant influenced by interpersonal mediation and social status demonstration. At present, China has not formed better low-carbon and green social norms. Under the influence of face consciousness, people tend to show their social status and income level by means of conspicuous and high consumption [15]. Therefore, strengthening pro-environmental education will be more conducive in the process of socialization.

#### 5.1.2. Descriptive Statistics Analysis of Behavior Intention of Energy Conservation

As for BIEC, most of the respondents had strong intentions for energy conservation. Compared to the descriptive statistics analysis of energy conservation, the survey results showed that most of the respondents were willing to change their energy use habits and purchase efficiency appliances, but they were not willing to sacrifice life quality. They were also most reluctant to implement IFB. On the whole, respondents had a strong BIEC when it was not necessary to spend a great deal of effort and they did not lose their own interests. There were inconsistencies between intention and QTB as well as between intention and IFB. People generally know how to reduce their energy use, but sometimes these motives of economics, convenience, and comfort would adjust, disturb, or even change behaviors [14]. 

#### 5.1.3. Descriptive Statistics Analysis of Perception about Energy-Conservation Results

As for PER, the respondents had better PERE than PERS. Some of them agreed with the viewpoint that "energy conservation can bring economic savings”, but some of them had negative or uncertain perceptions of the spiritual results. Overall, PER should be promoted. One of the reasons may have been because it is difficult for residents to obtain information and measure their amount of energy saving. If given feedback on energy conservation, there would be a significant reduction in energy use [45]. If given interpersonal comparisons, such as publicly setting energy efficiency goals, residents’ energy conservation would be effectively encouraged [45,46].

#### 5.1.4. Descriptive Statistics Analysis of Contextual Factors

With regard to contextual factors, most of the respondents believed that they were not good enough to be conducive to implementing energy conservation behavior. SNEC was not well received, which implies that China’s SNEC is not well established, which is consistent with the view of Mi et al. (2018) [15]. As for PEP and EVEP, energy conservation policies have failed to be well popularized and implemented. There was a certain proportion of respondents who could not understand energy conservation policies or perhaps were not familiar with ways to benefit from preferential policies. Further, guiding policies of energy conservation are not well popularized, such as “Public Energy Conservation Week” and the “Guide for Public Energy Conservation Behavior”. As for EVII, it is not easy to obtain related and useful information of energy conservation. The power sector of Jiangsu province has made a platform to obtain information about real-time electricity consumption since 2010. However, from the research results, users have failed to grasp or been indifferent toward how to use this system. As for EPA and PRIP, most of the respondents considered high scores of EPA, which shows that the market of energy saving products in China is well developed. However, respondents presented more sensitivity to the price of energy conservation products, especially energy efficient appliances. Motives of economics are the key for energy conservation behavior [14]. As for PRIE, most of the respondents rated it at a medium level with few effects on ECB.

### 5.2. Correlation Analysis of Variables

Correlation analysis between model variables is the basis for further testing of the model. Pearson correlation coefficient was used to test the correlation between independent variables and dependent variables, as well as the correlation between dependent variables and regulatory variables. Table 4 shows the relationship between ECB and its influencing factors. A correlation analysis of ECB and BIEC revealed that there were significant positive correlations between these variables. As for PER, it was significantly correlated with ECB and BIEC. For contextual factors, ECB was significantly correlated with SNEC, PEP, EVEP, EVII, and PRIE. EPA and PRIP were significantly correlated with EIB. On this basis, multiple linear regression analysis and hierarchical regression analysis were adopted to explore the effects of the CF and PER.

### 5.3. Modulatory Effect of the Contextual Factors

Hierarchical regression equations are often used to test the moderating effects of variables. According to the steps of moderating effect examination, the variables were first normalized, the models were built, and then the product terms of behavior and the contextual factors were constructed. Next, behavior, behavior intention, contextual factors, and product terms were incorporated into the equation step by step. Finally, if the regression coefficients of the product terms were significant, or the decision coefficients of the hierarchical regression model increased significantly, the modulatory effect of that contextual factor was considered to be significant. The results of the hierarchical regression analysis of the modulatory effects are shown in Table 5.

#### 5.3.1. Modulatory Effect on Intention to Habits-Adjustment Behavior

As shown in Table 5, the results of the multiple hierarchical regression analysis of modulatory effects showed that, in terms of the path of BIEC to HAB, the determination coefficients of model 3 were greater than those of models 1 and 2. Among the contextual variables, the positive modulatory effects of SNEC, PEP, EVEP, EVII, and PRIE were significantly present in the interaction between BIEC and HAB with positive effects. That is to say, if the SNEC was improved, households were more willing to do HAB. At the same time, if the related energy conservation policies could be better popularized and implemented, and more useful information and interventions were provided, BIEC could be more effectively transformed to HAB. In addition, the high PRIE could strengthen the transformation of intention to HAB.

#### 5.3.2. Modulatory Effect on Intention to Quality-Threshold Behavior 

The results of the multiple hierarchical regression analysis of modulatory effects (Table 5) showed that the determination coefficients of model 3 were greater than those of models 1 and 2 with regard to QTB. The modulatory effects of contextual variables (SNEC, PEP, EVEP, EVII, and PRIE) were significant and positive in the interaction between BIEC and QTB. The same was true for HAB in that an improved SNEC strengthened residents’ willingness to reduce their energy use, although it may have affected their quality of life. Meanwhile, in cases where PEP, EVEP, and EVII were better, BIEC could be transformed to QTB well. Additionally, high PRIE could also strengthen the transformation of BIEC to QTB.

#### 5.3.3. Modulatory Effect on Intention to Efficiency-Investment Behavior 

Based on the results of the multiple hierarchical regression analysis of modulatory effects as shown in Table 5, the determination coefficients of model 3 were greater than those of models 1 and 2 with regard to EIB. The modulatory effects of the contextual variables (PEP, EVEP, EVII, PRIE, EPA, and PRIP) were significant in the interaction between BIEC and EIB. Better PEP, EVEP, and EVII could strengthen residents’ willingness to EIB, although it required additional expenditures. Meanwhile, PRIE could also strengthen the transformation of BIEC to EIB. Under conditions of superior EPA, residents had stronger intentions and more behaviors to conserve energy by means of EIB. However, PRIP inhibited the transformation of intention to EIB. It was also notable that SNEC had no significant modulatory role on the path of BIEC to EIB.

#### 5.3.4. Modulatory Effect on Intention to Interpersonal-Facilitation Behavior 

The results of the multiple hierarchical regression analysis (Table 5) showed that, with regard to IFB, the determination coefficients of model 3 were greater than those of models 1 and 2. The modulatory effects of the contextual variables (SNEC, PEP, EVEP, and EVII) were significant and positive in the interaction between BIEC and IFB. Better PEP, EVEP, and EVII could strengthen residents’ willingness to promote others’ energy conservation. PRIE had no significant modulatory role in the path of BIEC to IFB, i.e., economic means had no significant effect on this path.

#### 5.3.5. Discussion of Modulatory Effect on Intention to Behavior

To sum up, the better the residents feel about SNEC, the more HAB, QTB, and IFB will be implemented. Energy conservation practitioners might develop effective programs to motivate individuals to engage in ECB from the perspective of SNEC [47,48]. Our study finds that SNEC can strengthen intentions to HAB, QTB, and IFB, but not to EIB. SNEC shaping should be from the different perspectives of promoting HAB, QTB, and IFB.

Existing studies have conveyed concerns about the impact of policy on residential energy consumption, public acceptance, and so on [49,50]. Our study mainly focuses on popularization, execution, and validity of energy conservation policy. From the analysis above, PEP and EVEP have significant regulatory effects on the path of intention to behavior, and although existing energy conservation policies are well established, their popularization, execution, and validity are not good enough. Policy making is important, but popularization, execution, and validity are more important.

Existing studies prove that information intervention is an effective way to encourage energy conservation [27,36,37], and these conclusions were drawn by experimentation. Actually, valid information of encouraging energy conservation cannot be obtained easily. Practitioners should pay more attention to EVII in consideration of its high effectiveness. 

PRIE is an effective measure to guide public ECB, but in China, electricity and natural gas are considered to be life necessities, and their prices have an impact on people’s livelihoods and are under the control of the Chinese government. In order to prevent the negative social impact of the adjustment of living energy prices, China has been trying to use a tiered pricing system. The practice shows that although the tiered pricing system has some effect (using tiered electricity price as an example), the main problems still persist, including inadequate levels of public awareness and acceptance, unreasonable tiered and electricity price standards, and insufficient reflection in regional differences [51].

EPA has significant positive moderate effects on intention to EIB, while PRIP has negative effects. Cost, quality, credibility, availability, and efficiency are the important factors that consumers consider the most when purchasing energy efficient appliances [52]. Therefore, EIB, EPA, and PRIP are the key measures, as well as PEP, EVEP, EVII, and PRIE.

### 5.4. Feedback Effects of Perception of Energy-Conservation Behavior Results

The method of linear regression analysis was used to test the effects of PER to BIEC and behavior. To avoid multi-collinearity among dimensions, variables were decentralized. According to the results of the linear regression analysis (Table 6), PERE showed a positive reinforcement of BIEC, and the regression coefficient was 0.078 under the significance level of 99.99%. As for ECB, PERE had positive reinforcement to HAB, QTB, and EIB, and their regression coefficients were 0.098, 0.135, and 0.146 under the significance level of 99.99%. The regression coefficients showed that the effect of PERE on EIB was greater than on others. However, PERE had no regressive relationship with IFB.

PERS had positive reinforcement to BIEC, and the regression coefficient was 0.293 under the significance level of 99.99%. PERS had positive reinforcement to HAB, QTB, and IFB, and their regression coefficients were 0.085, 0.103, and 0.144 under the significance level of 99.99%. The regression coefficients showed that the effect of PERE on IFB was larger than that of PERS on others. However, PERS had no regressive relationship with EIB. On the whole, BIEC and PERS had more significant and greater impacts than PERE. Because there were significant impacts of PER on BIEC and ECB, it is not only necessary to provide access to the information regarding the economic benefits of energy saving but also to promote spiritual satisfaction of ECB by means of persuasion and education. 

### 5.5. Hypothesis Test and Model Correction

On the basis of the empirical analysis results, the hypotheses could be tested. Hypothesis 1, i.e., BIEC has a positive effect on all of the four behaviors, passed the test. Hypothesis 2, i.e., contextual factors have significant modulatory effects on the path of BIEC to the four types of behavior, partially passed the test, because SNEC had an insignificant modulatory effect on EIB, and PRIE had an insignificant modulatory effect on IFB. Hypothesis 3, i.e., the two types of PER have significant impacts on BIEC, was totally significant and passed the test. Hypothesis 4, i.e., the two types of PER have significant impacts on the four types of behavior, partially passed the test because PERE had an insignificant effect on IFB, and PERS had an insignificant adjustment effect on EIB. Based on the empirical analysis conclusion, the conceptual framework was amended, as shown in Figure 2.

## 6. Conclusions and Policy Implications

In this paper, ECB was divided into four types, considering habit adjustment, quality threshold, efficiency investment, and interpersonal facilitation. A comprehensive conceptual framework was built, adding PBR and contextual factors from a new perspective. This study analyzed different effects of the intentions of energy conservation behavior on the four types of ECB through empirical tests, and further excavated the modulatory effects of contextual factors on the relationship between intention and behavior. The effect of perception of energy conservation results on intention and four types of behavior was examined in an empirical analysis. The conclusions and policy implications are as follows.

First, based on the empirical analysis of the investigation results, BIEC is the determinant of behavior, but BIEC does not well translate into behavior. An individual with high BIEC may not have high frequency of implementing the ECB. The degree of transformation of BIEC to different ECB is discrepant. Among the four types of ECB, QTB and IFB residents are most reluctant to act on QTB and IFB. Unless they are environmentalists or residents with high economic pressure, most of the public are reluctant to save energy at the expense of quality of life. Therefore, guiding ECB should pay more attention to the other types of behavior. As for IFB, China is a high-context country where people’s behavior is more influenced by others, but conspicuous consumption is still widely affecting people in our society without a good climate for energy conservation. Energy conservation has not yet become the daily topic concerning the public. A supportive environment for interpersonal facilitation is lacking. The policy should focus on shaping the mainstream consumption concept of green, pro-environmental, and energy conservation to be the new fashion. Platforms and opportunities for energy conservation communication should be provided; consequently, people will be more willing to take the initiative to exchange and promote energy conservation. More efforts should be made to publicize the knowledge and skills of energy conservation, thus residents can master more energy conservation skills and form habits of energy conservation in daily life. Practitioners should standardize the market for energy efficient products and guide people to give priority to energy efficient products when buying energy consuming products. 

Second, contextual factors have positive effects on the relationship between intention and behavior, but for different types of energy saving behaviors, the significance is different. SNEC has a positive effect on the relationship between BIEC and HAB. QTB, IFB. PEP, EVEP, and EVII can positively affect the relationship between BIEC and the four types of ECB. There are negative effects of PRIE and PRIP; PRIE has a negative effect on the relationship between BIEC and HAB, QTB, and EIB, while PRIP negatively affects the relationship between BIEC and EIB. The results of the empirical test imply that SNEC is not well received, and that policies of ECB are not well popularized or implemented. It is not easy to obtain related and useful information about energy conservation. To a great extent, modulating effects of contextual factors as amplifiers makes no effective difference. Therefore, it is urgent that it be promoted, and when designing specific contents of policies guiding ECB, it would be beneficial to consider these differences to ensure the effectiveness of the policy. Shaping SNEC, strengthening the PEP and EVEP, and enriching information intervention approaches will be more conducive to guiding the ECB of residents. Optimizing energy saving product attributes and rationalizing prices of energy and efficiency products will be also beneficial.

Finally, PERE has a positive impact on BIEC and HAB, QTB, EIB, while PERS positively affects HAB, EIB, and IFB. PERE has a larger incentive effect on BIEC than PERS, but the evaluations of PER are lower than BIEC. Optimizing the present contextual factors and thereby strengthening PER is particularly important for providing access to the information of economic benefits of energy saving and promoting satisfaction of energy conservation by means of persuasion and education.

## Figures and Tables

**Figure 1 ijerph-16-00939-f001:**
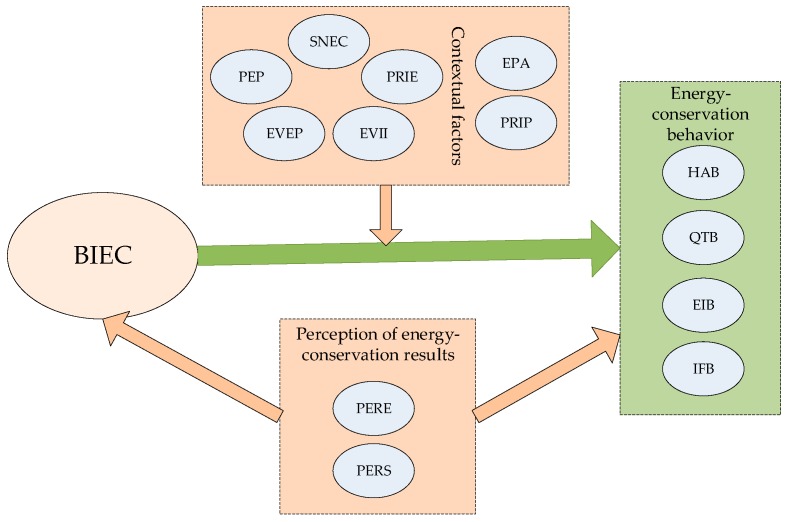
Conceptual framework of energy conservation behavior (ECB).

**Figure 2 ijerph-16-00939-f002:**
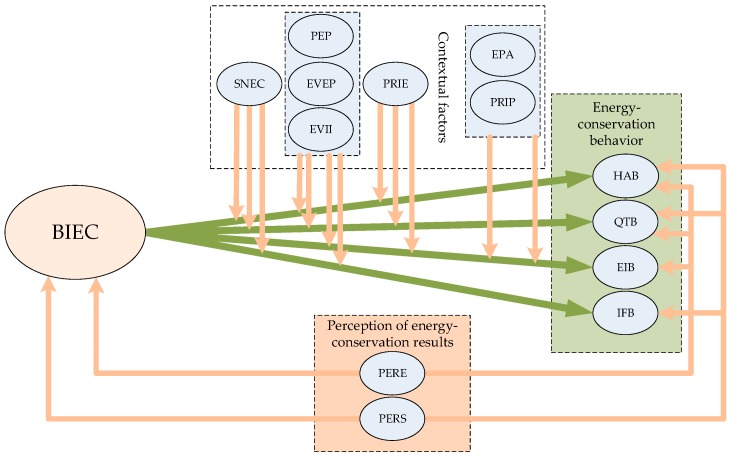
The modified conceptual framework of ECB.

**Table 1 ijerph-16-00939-t001:** Abbreviations and definitions of variables.

Variables		Abbreviations	Definitions
Energy conservation behavior (ECB)	Habit-adjustment behavior	HAB	The adjustment of behavior habits to reduce the use of energy without sacrificing the quality of life
	Quality-threshold behavior	QTB	Changes in the daily use of energy under conditions of sacrifice of a certain quality of life
	Efficiency-investment behavior	EIB	Reducing energy activities by investing in energy efficient products or equipment to improve energy efficiency
	Interpersonal-facilitation behavior	IFB	Promoting others’ energy conservation behavior through interpersonal activity
Behavior intention of energy conservation		BIEC	Individual’s intention to make efforts to implement energy conservation behaviors
Perception about energy conservation results (PER)	Perception about energy conservation results on economic savings	PERE	Perception about the economic savings achieved from the actual energy conservation behavior
	Perception about energy conservation results in spiritual satisfaction	PERS	Perception about the spiritual satisfaction achieved from the actual energy conservation behavior.
Contextual factors (CF)	Social norms of energy conservation	SNEC	The norms including the climate, the code of ethics, the state of public opinion and the code of conduct for energy conservation
	Popularization of energy conservation policy	PEP	The degree of residents’ familiarity to energy conservation policy
	Execution and validity of energy conservation policy	EVEP	The influencing degree of energy-conservation policy to residents’ energy conservation behavior
	Execution and validity of information intervention	EVII	The availability and usefulness of energy information
	Price of energy	PRIE	The price level of energy products used by residents in daily life, including electricity and gas prices
	Energy conservation product attributes	EPA	The attributes including the technical level, service quality, credibility and availability of energy conservation products
	Price of energy conservation products	PRIP	The price of efficiency household energy conservation appliances.

**Table 2 ijerph-16-00939-t002:** Distribution of social-demographic characteristics factors.

Variables	Item	Percent (%)	Variables	Item	Percent (%)
Gender	Male	48.85	Occupation	Worker	17.30
	Female	51.15		Civil servant	16.79
Age	18–28	29.24		Educator, scientist	13.22
	29–44	34.46		Private enterprise owners or employees	16.38
	45–59	28.39			
	Over 59	7.91		Individual business owners or employees	19.77
Education	Junior middle school or below	4.33		Retiree, the unemployed	10.85
				Others	5.69
	Senior middle school	18.45	Family income	Less than ¥3000	11.94
	Junior college	29.68		¥3001 to ¥5000	17.37
	Bachelor	32.71		¥5001 to ¥10,000	36.51
	Master and above	14.83		¥10,001 to ¥20,000	29.11
				Over ¥20,000	5.07

**Table 3 ijerph-16-00939-t003:** Results of scales test and descriptive statistics analysis.

Variables		Items (N)	Cronbach’s Alpha	KMO	Bartlett’s Test of Sphericity	Sig0.	Means	S0.D0.
ECB	HAB	4	0.860	0.894	98,300.451	0.000	30.42	10.168
	QTB	4	0.854			0.000	20.58	10.183
	EIB	3	0.816			0.000	20.99	10.265
	IFB	3	0.824			0.000	20.36	10.376
IECB		4	0.835	0.863	6920.774	0.000	30.25	00.959
PER	PERE	2	0.701	0.845	7560.238	0.000	30.04	00.912
	PERS	2	0.779			0.000	20.83	00.933
CF	SNEC	4	0.812	0.781	146,720.583	0.000	20.45	00.985
	PEP	4	0.824			0.000	20.66	10.371
	EVEP	2	0.769			0.000	20.32	10.293
	EVII	2	0.675			0.000	20.95	00.955
	EPA	4	00.842			00.000	30.21	00.941
	PRIE	2	00.881			00.000	30.46	00.927
	PRIP	2	00.843			00.000	30.95	00.916

**Table 4 ijerph-16-00939-t004:** Correlativity test of variables.

	HAB	QTB	EIB	IFB	BIEC	PER	SNEC	PEP	EVEP	EVII	PRIE	PRIP	EPA
**HAB**	1												
**QTB**	0.385 **	1											
**EIB**	0.326 **	0.438 **	1										
**IFB**	0.342 **	0.371 **	0.384**	1									
**BIEC**	0.393 **	0.246 **	0.317**	0.269 **	1								
**PER**	0.352 **	0.229 **	0.319 **	0.265 **	0.421 **	1							
**SNEC**	0.245 **	0.231 **	0.198 **	0.275 **	0.173 **	0.109 **	1						
**PEP**	0.134 **	0.117 **	0.294 **	0.177 **	0.151 **	0.147 **	0.134 **	1					
**EVEP**	0.084 **	0.076 **	0.114 **	0.189 **	0.132 **	0.165 **	0.147 **	0.143 **	1				
**EVII**	0.126 **	0.115 **	0.104 **	0.172 **	0.203 **	0.175 **	0.204 **	0.186 **	0.221 **	1			
**PRIE**	0.213 **	0.278 **	0.305 **	0.143 **	0.227 **	0.245 **	0.178 **	0.058 *	0.106 **	0.224 **	1		
**PRIP**	−0.045	−0.032	−0.186 **	0.013	−0.123 **	−0.257 **	−0.035	−0.078 *	0.041	0.015	0.018	1	
**EPA**	0.032	0.086 *	0.156 **	0.028	0.131 **	0.143 **	0.034	0.091 *	0.027	0.031	0.029	0.145 **	1

Note: * *p* < 0.05 level, ** *p* < 0.01 level.

**Table 5 ijerph-16-00939-t005:** Hierarchical regression analysis of the modulatory effects.

	**HAB**			**QTB**		
	**I**	**II**	**III**	**I**	**II**	**III**
**BIEC**	0.412 ***	0.385 ***	0.357 ***	0.406 **	0.371 **	0.344 **
**SNEC**		0.198 ***	0.112 ***		0.214 **	0.143 **
**PEP**		0.209 **	0.167 **		0.223 **	0.189 **
**EVEP**		0.215 **	0.181 **		0.176 ***	0.120 ***
**EVII**		0.210 **	0.165 **		0.241 ***	0.203 **
**PRIE**		0.186 ***	0.132 ***		0.183 ***	0.139 **
**BIEC** **×SNEC**			0.111 ***			0.074 **
**BIEC** **×PEP**			0.095 **			0.075 **
**BIEC** **×EVEP**			0.142 ***			0.099 **
**BIEC** **×EVII**			0.079 *			0.091 **
**BIEC** **×PRIE**			0.103 **			0.094 **
**R^2^**	0.255	0.283	0.339	0.210	0.243	0.285
**F-value**	4090.567	1950.249	1420.046	3450.864	2010.522	1230.554
	**EIB**			**IFB**		
	**I**	**II**	**III**	**I**	**II**	**III**
**BIEC**	0.391 ***	0.365 ***	0.330 ***	0.417 **	0.384 **	0.349 **
**SNEC**		0.145 ***	0.116 ***		0.184 ***	0.150 ***
**PEP**		0.165 ***	0.135 ***		0.257 **	0.216 **
**EVEP**		0.238 **	0.201 **		0.193 ***	0.154 **
**EVII**		0.221 **	0.186 **		0.218 **	0.183 **
**PRIE**		0.165 ***	0.124 **		0.204 **	0.168 **
**PRIP**		0.181 ***	0.132 **			
**EPA**		0.205 ***	0.173 **			
**BIEC** **×SNEC**			051			0.085 **
**BIEC** **×PEP**			0.117 ***			0.031
**BIEC** **×EVEP**			0.077 *			0.102 **
**BIEC** **×EVII**			0.082 *			0.113 **
**BIEC** **×PRIE**			0.105 **			0.044
**BIEC** **×EPA**			0.136 ***			
**BIEC** **×PRIP**			−0.154 ***			
**R^2^**	0.241	0.279	0.312	0.198	0.236	0.274
**F-value**	2470.236	1540.376	1070.854	2590.245	1740.698	1200.631

Note: * *p* < 0.05 level, ** *p* < 0.01 level, *** *p* < 0.001 level. R^2^ is the coefficient of determination.

**Table 6 ijerph-16-00939-t006:** Linear regression analysis of perception of energy conservation behavior results.

Dependent Variable	Independent Variable	Standardized Regression Coefficients	*t*-Value	Sig0.
BIEC	C		270.748	0.000
	PERE	0.078	40.673	0.000
	PERS	0.293	70.762	0.000
HAB	C		210.365	0.000
	PERE	0.098	40.804	0.000
	PERS	0.085	40.703	0.000
QTB	C		250.001	0.000
	PERE	0.135	50.955	0.000
	PERS	0.103	40.909	0.000
EIB	C		160.064	0.000
	PERE	0.146	60.257	0.000
	PERS	0.013	10.053	0.060
IFB	C		170.634	0.000
	PERE	0.042	10.178	0.033
	PERS	0.144	50.931	0.000

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
