# Peer review of "Empirical Study on Households’ Energy-Conservation Behavior of Jiangsu Province in China: The Role of Policies and Behavior Results"

_ijerph, 2019, doi:10.3390/ijerph16060939_

Reviewer 1 Report

Thank you for the opportunity to review the manuscript.

The study identifies four energy conservation behaviors and modeled the determinants of these behaviors. Clearly, the study has promise and could make an important contribution to the scholarship. However, the paper will need substantial contribution to be ready.

First, Section 3, the hypotheses and conceptual framework, is very underdeveloped. The authors identify several theories that form the foundation of the paper without explicating on each of them. They wrote:

On the basis of the Theory of Planned Behavior [30], the Responsible Environmental Behavior Model [53], and the Theory of Interpersonal Behavior [54], the behavior of an individual is not only affected by psychological characteristics, but it is also influenced by the surrounding environment and other individuals’ behavior.”

 I suggest the authors review relevant literature on these theories and develop their testable hypotheses from these. Readers will appreciate information on the assumptions of these theories and how they are being operationalized in the study.

Second, there is weak link between the various parts of the paper. The results should be presented as to tell the reader whether the theory/ hypotheses are supported or not. As it is presently, the parts of the paper are disjointed. Also in the conclusion, the authors must demonstrate whether the theories are supported or not and the implication and contribution of such a finding. It will be helpful to include a discussion on the implication of the paper. There should be a few take ways from the paper to the reader. As it is, the novel contribution have not been clearly identified. The authors thus miss an opportunity to show the relevance of their study.

Third, the authors stated several socio-demographic variables such as age, gender, education, income, etc. I expected that these controls would be tested. Absent these controls, the results are not convincing.

Fourth, I suggest the authors edit the paper to improve style and readability.

Author Response

Manuscript Number: ijerph-438419

Article Title: Empirical Study of Households’ Energy-conservation Behavior: The Role of Policies and Behavior Results

Responses to reviewers are listed below in red.

Reviewer 1

Reviewer #1: Thank you for the opportunity to review the manuscript.

The study identifies four energy conservation behaviors and modeled the determinants of these behaviors. Clearly, the study has promise and could make an important contribution to the scholarship.

Response: Thank you for your positive comment.

1. However, the paper will need substantial contribution to be ready. First, Section 3, the hypotheses and conceptual framework, is very underdeveloped. The authors identify several theories that form the foundation of the paper without explicating on each of them. They wrote: “On the basis of the Theory of Planned Behavior [30], the Responsible Environmental Behavior Model [53], and the Theory of Interpersonal Behavior [54], the behavior of an individual is not only affected by psychological characteristics, but it is also influenced by the surrounding environment and other individuals’ behavior.” I suggest the authors review relevant literature on these theories and develop their testable hypotheses from these. Readers will appreciate information on the assumptions of these theories and how they are being operationalized in the study.

Response: Thank you for your valuable comments. Based on your suggestions, we have supplemented the relevant literature on the Theory of Planned Behavior, the Theory of Responsible Environmental Behavior, and the Theory of Interpersonal Behavior. We also further explained the basis of the hypotheses. (See page 3, Section 3.1, line 109-118)

2. Second, there is weak link between the various parts of the paper. The results should be presented as to tell the reader whether the theory/ hypotheses are supported or not. As it is presently, the parts of the paper are disjointed. Also in the conclusion, the authors must demonstrate whether the theories are supported or not and the implication and contribution of such a finding. It will be helpful to include a discussion on the implication of the paper. There should be a few take ways from the paper to the reader. As it is, the novel contribution have not been clearly identified. The authors thus miss an opportunity to show the relevance of their study.

Response: Thank you for your valuable comments and kind suggestions. Based on your suggestions, we have further strengthened the relations among all the chapters and supplemented hypothesis test in the “5.5 Hypothesis test and model correction”. We also have further supplemented the discussion of the paper in Section 5. We have supplemented discussions following results of statistics analysis and added Section 5.3.5 to discuss modulatory effect.See pages 7-13

3. Third, the authors stated several socio-demographic variables such as age, gender, education, income, etc. I expected that these controls would be tested. Absent these controls, the results are not convincing.

Response: Thank you for your valuable comments. In the Theory of Planned Behavior (Ajzen, 1991), the socio-demographic characteristic variable is considered as one of the antecedent variables of energy conservation intention. And scholars usually make difference analysis on social demographic characteristics (Jain SK & Kaur, 2006; Frederiks et al., 2015). This paper mainly focuses on the modulatory effects of situational factors in the transition from intention to behavior and the effect of behavioral outcome on intention and behavior after the implementation of behavior. The theoretical framework in this paper does not include the path to analyze the effect of social demographic variables on intention and behavior. Here, only descriptive statistical analysis is performed for socio-demographic variables to analyze whether the questionnaire is evenly distributed in terms of demographic characteristics and whether it meets the requirements of social investigation. Therefore, this paper did not test and analyze the effect of social-demographic variables on energy-conservation intention and behavior.

Ajzen I. The Theory of Planned Behavior. Organizational Behavior and Human Decision Processes, 1991, 50(2): 179-211.

Jain SK, Kaur G. Role of socio-demographics in segmenting and profiling green consumers: an exploratory study of consumers in India. Journal of International Consumer Marketing, 2006, 18:3, 107-146.

Frederiks RE, Stenner K, Hobman VE. The socio-demographic and psychological predictors of residential energy consumption: a comprehensive review. Energies, 2015, 8, 573-609.

4. Fourth, I suggest the authors edit the paper to improve style and readability.

Response: Thank you for your valuable suggestions. This paper has been edited by professional language polishing company. We have further edited the whole revision to improve style and readability with the help of an English native speaker. 

Reviewer 2 Report

Overall

Even though the paper is well written, the subject is quite interesting and the bibliography is rich, it is too long and confused in the variable definition. It must be definitively shortened and all the variables used in the text must be well defined. Tables and Figures must be self-explained: please, detail more.

Article title

Since the paper is based on a case study in the Jiangsu province, it must be highlighted it in the title.

1. Introduction

Introduction can be shortened.

References [1] and [2] are specific for the China, while you in the text refer to ‘global warming’, that I suppose involving all the world.

Reference [5] similarly refers to ‘global emissions’ while it is the result of a Japanese household survey. Again: here you write of a 11% global emission, while at line 42 there is a 17% for the U.S. Please specify better.

Line 69-70: ‘However, there are usually gaps between intention and energy-conservation behavior, and especially under the influence of China’s background of policies [17].’ …. But [17] is: Attitude–behavior gap in energy issues: Case study of three different Finnish residential areas. Please clarify.

2. Literature review

Interesting, but it sharply necessary to shorten this chapter. The literary review is very rich, but you did not use it for the discussion.

Line 120: Many scholars … I see only 2 references.

3. Conceptual framework

Line 189: perhaps the reference is [30] instead of [31].

Line 196: ‘Perception about energy-conservation results (PER) contain economic savings dimensions (ESD) and spiritual satisfaction dimensions (SSD). The conceptual framework is shown as Fig 1.’. I don’t find PER, ESD and SSD in figure 1, neither in the text.

Figure 1: What does PEBE and PEBS mean? May you explain them in the text? These terms are often used in the paper, but they were never defined.

4. Methods

Line 227: ‘The formal investigation began in June 1, 2018 in Jiangsu province of China. Until February 25, 2018, a total of 236 paper questionnaires and 478 network questionnaires were recycled.’ I don’t understand: you started the investigation in June, 2018 and collected the questionnaires in February, 2018?

Table 1. If you use female, please use male (and not man).

Table 1. I don’t understand the symbol Ұ. Please remember that this an international journal.

Results

Table 2: What does it meas Cronbach’sa?

IECB: what is this? Never defined

PEB, PEBE, PEBS: never defined.

PEA: never defined

Table 3: BIEC: never defined

FEP: never defined

*** Statistical significance: not present in the table

I can’t continue to read the results, because too many variables were never defined and I can’t appreciate your comments.

Discussion

You called chapter 5 ‘Results and discussion’, but I don’t see any type of discussion and comparison with other similar works (and in the reference you put a lot of them…).

Please, add a real discussion.

Conclusion

The first sentence is not understandable.

Author Response

Manuscript Number: ijerph-438419

Article Title: Empirical Study of Households’ Energy-conservation Behavior: The Role of Policies and Behavior Results

Responses to reviewers are listed below in red.

Reviewer 2

Reviewer #2: Overall

Even though the paper is well written, the subject is quite interesting and the bibliography is rich, it is too long and confused in the variable definition. It must be definitively shortened and all the variables used in the text must be well defined. Tables and Figures must be self-explained: please, detail more.

Response: Thank you for your valuable comments. We have deleted some contents to shorten the article. All of the variables used in the text are put in a table to show their definitions and abbreviations. (See page 3-4, Table 1)

Article title

Since the paper is based on a case study in the Jiangsu province, it must be highlighted it in the title.

Response: Thank you for your valuable suggestions. Based on your suggestions, we have supplemented the theme of Jiangsu province. And the new title is “Empirical Study of Households’ Energy-conservation Behavior of Jiangsu Province in China: The Role of Policies and Behavior Results”.

1. Introduction

(1) Introduction can be shortened.

Response: Thank you for your valuable comments. We have deleted some contents to shorten the introduction.

(2) References [1] and [2] are specific for the China, while you in the text refer to ‘global warming’, that I suppose involving all the world.

Response: Thank you for your valuable comments. As to “References [1] and [2]”, “global warming” here refers to one of the problems caused by energy consumption, i.e., global climate change. As is known, China contributes the largest total emission of carbon dioxide (CO2) and sulfur dioxide (SO2), which causes serious climate change. Globally, a reduction of energy consumption is urgently needed to limit climate change, especially in China. “Global warming” may not be very suitable and we have revised it into “climate change”. From references [1] and [2], we can find that the authors have the viewpoint of energy consumption can cause the climate change, and this problem is severe especially for China.

…“It is well known that carbon dioxide emissions are the main cause of global warming. … Therefore, in order to protect the environment and public health, carbon emissions need to be reduced as soon as possible. The carbon emission intensity is most closely related to energy consumption.”…

[1]  Wei, J., Chen, H., Long, R. Determining multi-layer factors that drive the carbon capability of urban residents in response to climate change: An exploratory qualitative study in China. Int. J. Environ. Res. Public Health. 2018, 15: 1607.

…“As the largest developing country in the world, China has experienced 30 years of rapid growth, and through reforms and more open access, it has become the world’s second largest economy. However, China’s rapid development has also consumed large amounts of energy and resources, and has brought on a series of environmental problems. For example, globally, China contributes the largest total emission of carbon dioxide (CO2) and sulfur dioxide (SO2). ” …

[2]  Wang, Q.W., Zhao. Z.Y., Shen, N., et al. Have Chinese cities achieved the win–win between environmental protection and economic development? From the perspective of environmental efficiency. Ecol. Indic. 2015, 51: 151-8.

(3) Reference [5] similarly refers to ‘global emissions’ while it is the result of a Japanese household survey. Again: here you write of a 11% global emission, while at line 42 there is a 17% for the U.S. Please specify better.

Response: Thank you for your valuable comments. Here, we attempt to show the household energy consumption is significant for reduction. And we have revised them and reorganized the statement to state this viewpoint. (See Section 1, line 38-44)

(4) Line 69-70: ‘However, there are usually gaps between intention and energy-conservation behavior, and especially under the influence of China’s background of policies [17].’ …. But [17] is: Attitude–behavior gap in energy issues: Case study of three different Finnish residential areas. Please clarify.

Response: Thank you for your valuable comments. We have corrected the literatures’ position and supplemented the missed literature. (See Section 1, line 47-53)

Geng, J.C., Long, R.Y., Chen, H., et al. Exploring the motivation-behavior gap in urban residents’ green travel behavior: A theoretical and empirical study. Resour. Conserv. Recy. 2017, 125: 282-292.

2. Literature review

(1) Interesting, but it sharply necessary to shorten this chapter. The literary review is very rich, but you did not use it for the discussion.

Response: Thank you for your valuable comments and kind suggestions. We have deleted some contents to shorten the introduction. (See Section 2)

(2) Line 120: Many scholars … I see only 2 references.

Response: Thank you for your valuable comments. We have supplemented the related literatures and reorganized the statement. (See Section 2, line 82-94)

3. Conceptual framework

(1) Line 189: perhaps the reference is [30] instead of [31].

Response: Thank you for your valuable comments. We have checked all of the serial numbers and corrected this mistake.

(2) Line 196: ‘Perception about energy-conservation results (PER) contain economic savings dimensions (ESD) and spiritual satisfaction dimensions (SSD). The conceptual framework is shown as Fig 1.’. I don’t find PER, ESD and SSD in figure 1, neither in the text. Figure 1: What does PEBE and PEBS mean? May you explain them in the text? These terms are often used in the paper, but they were never defined.

Response: Thank you for your valuable comments. We have checked all of the abbreviations and ensured the consistent of the variables and their abbreviations in the context. All of the variables used in the text are put in a table to show their definitions and abbreviations. (See page 3-4, Table 1)

4. Methods

(1) Line 227: ‘The formal investigation began in June 1, 2018 in Jiangsu province of China. Until February 25, 2018, a total of 236 paper questionnaires and 478 network questionnaires were recycled.’ I don’t understand: you started the investigation in June, 2018 and collected the questionnaires in February, 2018?

Response: Thank you for your valuable comments. We are sorry for this clerical error and we have corrected it.

(2) Table 1. If you use female, please use male (and not man).

Table 1. I don’t understand the symbol Ұ. Please remember that this an international journal.

Response: Thank you for your scrupulous and valuable comments. We have revised the non-standard form of expression.

5. Results

(1) Table 2: What does it means Cronbach’s a?

Response: Thank you for your valuable comments. It is the abbreviation for Cronbach’s alpha which is used to measure reliability in statistics. It may be not displayed clearly because of formula editor. We have revised the abbreviation into the complete one.

(2)IECB: what is this? Never defined

PEB, PEBE, PEBS: never defined.

PEA: never defined

Table 3: BIEC: never defined

FEP: never defined

Response: Thank you for your valuable comments. We have checked all of the abbreviations and ensured the consistent of the variables and their abbreviations in the context. All of the variables used in the text are put in a table to show their definitions and abbreviations. (See page 3-4, Table 1)

(3) *** Statistical significance: not present in the table

Response: Thank you for your valuable comments. We have supplemented it in table note.

(4) I can’t continue to read the results, because too many variables were never defined and I can’t appreciate your comments.

Response: We are sorry for the abbreviations variables and thank you for your valuable comments. We have checked all of the abbreviations and ensured the consistent of the variables and their abbreviations in the context. All of the variables used in the text are put in a table to show their definitions and abbreviations. (See page 3-4, Table 1)

6. Discussion

You called chapter 5 ‘Results and discussion’, but I don’t see any type of discussion and comparison with other similar works (and in the reference you put a lot of them…).

Please, add a real discussion.

Response: Thank you for your valuable comments. We have further supplemented the discussion of the paper and compared other similar works in Section 5. We have supplemented discussions following results of statistics analysis and added Section 5.3.5 to discuss modulatory effect.See Section 5, pages 6-12

7. Conclusion

The first sentence is not understandable.

Response: Thank you for your valuable comments. We have re-edited and re-organized this sentence. And we also have further checked the whole article sentences to make sure it is readable.

Reviewer 3 Report

This paper describes the findings of an empirical study that addresses residential energy conservation behavior. While it also addresses energy conservation intention, and energy conservation behavior, it has a special focus on perception of energy conservation results and contextual factors.

The paper starts with an introduction motivating the topic by discussing the problems of greenhouse gas emissions and how relevant the private households’ energy consumption behavior is in that respect. The authors then present their review of existing literature where they point out factors that have been proven to influence energy conservation behavior. The review is followed by the presentation of the study hypotheses and the conceptual framework. To address the research questions the authors used a questionnaire that has been distributed online and offline resulting in a sample of 568 participants. After detailly presenting the results of the statistical analyses, the authors shortly conclude insights from the study.

********

The topic of the article is basically very interesting, and it is of great relevance to counteract the problem of greenhouse gas emission. However, the paper has in its current state some weaknesses that I will explain in more detail:

My biggest concern is the paper's overall contribution to the existing discourse. In the Literature Review section, the authors mention many studies that have already provided considerable results on questions relating to energy conservation intention, and energy conservation and perception or control of energy conservation results. In this respect, it does not become clear what new aspect will be added and thus make a contribution that goes beyond the findings of the studies mentioned above (and many other studies in this field of research). Or the authors do not describe their contribution clearly enough and do not clearly distinguish their research from the research of other studies. As also contextual factors have been studied before – the authors need to better work out, what is different in their study, if or in how far their contextual factors differ from those in other studies and why this might be necessary to investigate. What reinforces the impression of a low contribution is that in the discussion section the findings of the study at hand are not reflected against the background of the existing literature - here, too, it is not clear what exactly the added value or the extension of knowledge is. In how far do your results differ for example from source (29)?

My suggestion would be that the authors work out better how their study contributes to the existing discourse. They could do this, for example, by strengthening their argument that China is of particular interest as it strongly differs from other countries. So far, the authors mention only in the introductory chapters that there are differences to other cultures - but these are not elaborated further. In their first sections the authors sometimes shortly refer to the specific situation or context of China (“Face culture” or „especially under the influence of China’s background of policies“ à What is special about it?  Please do not expect every reader to know what you are pointing to). The authors miss the chance to work out why China is particularly important to consider. While they do explain, that China developed very quickly thus a huge increase in energy is expected, the authors fail to present how culture and policies differ from other countries where studies have already examined energy conservation behavior.

********

Another formal point that does not contribute to the good readability of the text is the inconsistent use of abbreviations.

·         Figure 1/Table 2: Inconsistent taxonomy: PEB, PEBE and PEBS are not introduced before, there is only PER (table2) and ESD (text, 197)?

·         Line 266 Is IECB also differentiated by the same four items than ECB? To what figures are you referring to at that point? That is very confusing.

·         The abbreviation PEA is also not introduced, or it might be mixed up with EPA?

·         In Table 3 you mix up abbreviations again (BIEC vs. IECB?)

·         Maybe a key could help to keep track with the abbreviations.

********

Other aspects in the order of their appearance in the text:

Abstract:

This goes in line with the conclusion; The authors only present statistical findings but what would be much more interesting are their meanings, recommended actions and what the central contribution of the paper is.

Introduction:

The introduction is redundant in some parts (line 64-67). There are central studies such as that of Stern (2000) and Selvefors et al (2015) that have not been discussed although they give central insights in contextual factors.

Further, there are some formulations which I did not quite understand.

·         What is meant by „the marginal effect“ (line 63)

·         The argument in this sentence is not clear to me:  “This approach is based on evidence that psychological factors (e.g., environmental awareness and concern) have significant effects on energy conservation behaviors; there is a gap between intention and behavior in energy conservation  (Line 111-113)

·         Please explain „face-culture“ shortly.

Literature review

The Literatur Review Part is in some parts confusing and I am not sure if this is also due to imperfect Grammar. Maybe rechecking and reformulating with a native speaker might help to fix that  (i.e. Line 149: “Behavior that has been happened has a significant influence on behavior intention, and the behavior results have residual effects”)

Hypotheses:

According to the literature review, all hypotheses have already been examined once, or there are studies on them. If the authors refer to specific relationships, that differ from those of studies before, they must better work out what the differences are. H1: In line 70 the authors wrote „The modulatory role of contextual factors and the subsequent effect of the perception of energy conservation results play the role in following behaviors“ and refer to source 18. As far as I understood with H1 you want to test exactly the same: Contextual factors have a significant modulatory effect on the path of intention to behavior. Same is with H2 and H3 and l. 70, l. 99, l. 151. Thus, it is not clear to me, what the gap is. Is it more a comparative or confirmatory analysis?

Hypothesis do not reflect all paths referred to in section 3.2

The provision of the survey and data in the supplementary sources might help reviewers and readers to comprehend the results.

Methods:

The authors do mention that they use Likert 5 method for their questionnaire, but further explanation is needed as to how the scale was labeled in order to fully comprehend the results presented in section 5.1

Results and discussion

The results section is relatively hard to follow. The authors do not use full terms or abbreviations consistently which creates confusion at some points. I.e. the authors should make it more transparent to what figure they refer i.e. by adding the figure in brackets (..393**).  What is noticeable when looking at the statistical correlations is that non of the correlations is higher than .5 which does not imply a strong relationship even if it is significant.

Only the results are described in the results and discussion section. The interpretation and the meaning of the results as well as the placing of the results in the overall discourse are not discussed enough in the discussion.

Others:

·         In 4.3.3. there is a typo or a mistake as you have the same starting sentence ending with „…with regard to quality-threshold behavior”.

·         No figures for the linear regression described in 5.4.

Conclusion and policy implications

Also, in the conclusion the authors do not add further discussion about modulatory role of contextual factors. There is only one sentence that goes a tiny bit beyond the purely descriptive reproduction of the results: “: On the whole, households have strong intention, less actual behavior and poorer perception, due to the modulatory effect of contextual factors that, to a great extent, do not function effectively”. (line 407). What do the authors mean by “that do not function effectively”? Is there some background information they are referring to? How are contextual factors in China? In what terms do they not work efficiently? How could this be changed? This needs to be discussed in more detail.

Furthermore, I am not an expert in statistical analyses, but I doubt the following statement is true: “present contextual factors weaken the intention of energy conservation.” (Line 418) as the interaction terms are positive, thus strengthen the positive relation?

There is only one sentence per implication which I find is not enough as those sentences are rather superficial and do not seem to be rooted in the results. What do the authors mean by “downstream links”? line 426? Please elaborate.

*******

Line spacing is not consistent.

Further research:

Sütterlin, Bernadette, Thomas A. Brunner, and Michael Siegrist. "Who puts the most energy into energy conservation? A segmentation of energy consumers based on energy-related behavioral characteristics." Energy Policy 39.12 (2011): 8137-8152.

Black, J. S., Stern, P. C., & Elworth, J. T. (1985). Personal and contextual influences  on household energy adaptations. Journal of Applied Psychology, 70, 3-21.

Selvefors, Anneli, I. C. Karlsson, and Ulrike Rahe. "Conflicts in everyday life: The influence of competing goals on domestic energy conservation." Sustainability 7.5 (2015): 5963-5980.

*******

I find the paper is not yet ready for publication in its current status and needs substantial Revision.

Author Response

Manuscript Number: ijerph-438419

Article Title: Empirical Study of Households’ Energy-conservation Behavior: The Role of Policies and Behavior Results

Responses to reviewers are listed below in red.

Reviewer 3

Reviewer #3: This paper describes the findings of an empirical study that addresses residential energy conservation behavior. While it also addresses energy conservation intention, and energy conservation behavior, it has a special focus on perception of energy conservation results and contextual factors.

The paper starts with an introduction motivating the topic by discussing the problems of greenhouse gas emissions and how relevant the private households’ energy consumption behavior is in that respect. The authors then present their review of existing literature where they point out factors that have been proven to influence energy conservation behavior. The review is followed by the presentation of the study hypotheses and the conceptual framework. To address the research questions the authors used a questionnaire that has been distributed online and offline resulting in a sample of 568 participants. After detailly presenting the results of the statistical analyses, the authors shortly conclude insights from the study.

Response: Thank you for your positive comment.

********

1. The topic of the article is basically very interesting, and it is of great relevance to counteract the problem of greenhouse gas emission. However, the paper has in its current state some weaknesses that I will explain in more detail:

My biggest concern is the paper's overall contribution to the existing discourse. In the Literature Review section, the authors mention many studies that have already provided considerable results on questions relating to energy conservation intention, and energy conservation and perception or control of energy conservation results. In this respect, it does not become clear what new aspect will be added and thus make a contribution that goes beyond the findings of the studies mentioned above (and many other studies in this field of research). Or the authors do not describe their contribution clearly enough and do not clearly distinguish their research from the research of other studies. As also contextual factors have been studied before – the authors need to better work out, what is different in their study, if or in how far their contextual factors differ from those in other studies and why this might be necessary to investigate. What reinforces the impression of a low contribution is that in the discussion section the findings of the study at hand are not reflected against the background of the existing literature - here, too, it is not clear what exactly the added value or the extension of knowledge is. In how far do your results differ for example from source (29)?

My suggestion would be that the authors work out better how their study contributes to the existing discourse. They could do this, for example, by strengthening their argument that China is of particular interest as it strongly differs from other countries. So far, the authors mention only in the introductory chapters that there are differences to other cultures - but these are not elaborated further. In their first sections the authors sometimes shortly refer to the specific situation or context of China (“Face culture” or, especially under the influence of China’s background of policies“ à What is special about it? Please do not expect every reader to know what you are pointing to). The authors miss the chance to work out why China is particularly important to consider. While they do explain, that China developed very quickly thus a huge increase in energy is expected, the authors fail to present how culture and policies differ from other countries where studies have already examined energy conservation behavior.

Response: Thank you for your valuable comments and kind suggestions. We have major revised the manuscript and supplemented some contexts in Introduction and Literature review to explain the purpose and contribution of our study and the differences from other studies.

2. Another formal point that does not contribute to the good readability of the text is the inconsistent use of abbreviations.

•Figure 1/Table 2: Inconsistent taxonomy: PEB, PEBE and PEBS are not introduced before, there is only PER (table2) and ESD (text, 197)?

•Line 266 Is IECB also differentiated by the same four items than ECB? To what figures are you referring to at that point? That is very confusing.

•The abbreviation PEA is also not introduced, or it might be mixed up with EPA?

•In Table 3 you mix up abbreviations again (BIEC vs. IECB?)不一致

•Maybe a key could help to keep track with the abbreviations.

Response: We are sorry for the abbreviations variables and thank you for your valuable comments. We have checked all of the abbreviations and ensured the consistent of the variables and their abbreviations in the context. All of the variables used in the text are put in a table to show their definition and abbreviations. (See page 3-4, Table 1)

3. Other aspects in the order of their appearance in the text:

(1) Abstract:

This goes in line with the conclusion; The authors only present statistical findings but what would be much more interesting are their meanings, recommended actions and what the central contribution of the paper is.                                                                        Response: Thank you for your valuable comments. We have revised the Abstract and supplemented the context of meanings and contribution.

(2) Introduction:

The introduction is redundant in some parts (line 64-67). There are central studies such as that of Stern (2000) and Selvefors et al (2015) that have not been discussed although they give central insights in contextual factors.

Response: Thank you for your valuable comments. We have deleted some redundant contents and added the valuable studies, such as that of Stern (2000), Selvefors et al. (2015) and Sütterlin et al. (2011).

(3) Further, there are some formulations which I did not quite understand.

•What is meant by “the marginal effect ”(line 63)

•The argument in this sentence is not clear to me: “This approach is based on evidence that psychological factors (e.g., environmental awareness and concern) have significant effects on energy conservation behaviors; there is a gap between intention and behavior in energy conservation (Line 111-113)

•Please explain, face-culture shortly.

Response: Thank you for your valuable comments. We have deleted the unclear contexts and sentences and reorganized the statement.

“Face-culture” means “China has a special background of ‘high-context culture’. The face sometimes means individual’s public image in society, which is concerned to be the social status and reputation. Face consciousness and conspicuous consumption usually results in high consumption [15,16].”

Mi, L.Y., Xue, Y., Yang, J., et al. Influence of conspicuous consumption motivation on high-carbon consumption behavior of residents—An empirical case study of Jiangsu province, China. J. Clean. Prod. 2018, 191: 167-81.

Wang, J.G., Wang, J.M., Du, Y. A review of the research on the attitude behavior gap in green consumption and future prospects. Collected Essays Finan. Econ. 2017, 11: 95-103. (In Chinese)

(4) Literature review

The Literature Review Part is in some parts confusing and I am not sure if this is also due to imperfect Grammar. Maybe rechecking and reformulating with a native speaker might help to fix that (i.e. Line 149: “Behavior that has been happened has a significant influence on behavior intention, and the behavior results have residual effects”)

Response: Thank you for your valuable comments. We have deleted the unclear contexts and sentences and reorganized the statement. And we have re-checked and re-edited the statement of the whole manuscript with the help of an English native speaker.

(5) Hypotheses:

According to the literature review, all hypotheses have already been examined once, or there are studies on them. If the authors refer to specific relationships, that differ from those of studies before, they must better work out what the differences are. H1: In line 70 the authors wrote, The modulatory role of contextual factors and the subsequent effect of the perception of energy conservation results play the role in following behaviors and refer to source 18. As far as I understood with H1 you want to test exactly the same: Contextual factors have a significant modulatory effect on the path of intention to behavior. Same is with H2 and H3 and l. 70, l. 99, l. 151. Thus, it is not clear to me, what the gap is. Is it more a comparative or confirmatory analysis?

Hypothesis do not reflect all paths referred to in section 3.2

The provision of the survey and data in the supplementary sources might help reviewers and readers to comprehend the results.

Response: Thank you for your valuable comments. We have supplemented the hypothesis and major revised the whole manuscript.

(6) Methods:

The authors do mention that they use Likert 5 method for their questionnaire, but further explanation is needed as to how the scale was labeled in order to fully comprehend the results presented in section 5.1

Response: Thank you for your valuable comments. We have supplemented an example to illustrate how to label the scale. (See line 175-177)

(7) Results and discussion

The results section is relatively hard to follow. The authors do not use full terms or abbreviations consistently which creates confusion at some points. I.e. the authors should make it more transparent to what figure they refer i.e. by adding the figure in brackets (..393**).  What is noticeable when looking at the statistical correlations is that non of the correlations is higher than .5 which does not imply a strong relationship even if it is significant.

Only the results are described in the results and discussion section. The interpretation and the meaning of the results as well as the placing of the results in the overall discourse are not discussed enough in the discussion.

Response: Thank you for your valuable comments. We have rechecked the whole manuscript and illustrated the variables. All the variables used in the text are put in a table to show their definitions and abbreviations. (See page 3, Table 1)

We have supplemented the “Note: *P < 0.05 level, **P < 0.01 level, ***P < 0.001 level” and revised the improper note.

We have further supplemented the discussion of the paper and compared other similar works in Section 5. We have supplemented discussions following results of statistics analysis and added Section 5.3.5 to discuss modulatory effect.See pages 6-12

(8) Others:

•In 4.3.3. there is a typo or a mistake as you have the same starting sentence ending with “…with regard to quality-threshold behavior”.

•No figures for the linear regression described in 5.4.

Response: Thank you for your valuable comments. We have re-checked and re-edited the statement of the whole manuscript. And we have supplemented the linear regression results in Table 6.

(9) Conclusion and policy implications

Also, in the conclusion the authors do not add further discussion about modulatory role of contextual factors. There is only one sentence that goes a tiny bit beyond the purely descriptive reproduction of the results: “On the whole, households have strong intention, less actual behavior and poorer perception, due to the modulatory effect of contextual factors that, to a great extent, do not function effectively”. (line 407). What do the authors mean by “that do not function effectively”? Is there some background information they are referring to? How are contextual factors in China? In what terms do they not work efficiently? How could this be changed? This needs to be discussed in more detail.

Furthermore, I am not an expert in statistical analyses, but I doubt the following statement is true: “present contextual factors weaken the intention of energy conservation.” (Line 418) as the interaction terms are positive, thus strengthen the positive relation?

There is only one sentence per implication which I find is not enough as those sentences are rather superficial and do not seem to be rooted in the results. What do the authors mean by “downstream links”? line 426? Please elaborate.

Response: Thank you for your valuable comments and kind suggestions. We have major revised the conclusion. (See Section 6. Conclusions and policy implications, Page 12-13)

(10) Line spacing is not consistent.

Response: Thank you for your valuable comments. We have normalized the article format.

Further research:

Sütterlin, Bernadette, Thomas A. Brunner, and Michael Siegrist. "Who puts the most energy into energy conservation? A segmentation of energy consumers based on energy-related behavioral characteristics." Energy Policy 39.12 (2011): 8137-8152.

Black, J. S., Stern, P. C., & Elworth, J. T. (1985). Personal and contextual influences  on household energy adaptations. Journal of Applied Psychology, 70, 3-21.

Selvefors, Anneli, I. C. Karlsson, and Ulrike Rahe. "Conflicts in everyday life: The influence of competing goals on domestic energy conservation." Sustainability 7.5 (2015): 5963-5980.

Response: Thank you for your valuable suggestions. We have downloaded these articles and cited them.

I find the paper is not yet ready for publication in its current status and needs substantial Revision.

Response: Thank you for your valuable comments. We have major revised this manuscript based your comments and advices. They are very meaningful and useful to improve the quality of our study. Thanks a lot again.

Round  2

Reviewer 1 Report

It was nice reading the paper one more time and appreciate that all my concerns have been addressed. 

Reviewer 2 Report

You improved a lot your work. Only one little typo error in Figure 1: PERB (never defined and never used).